# Stability and Removal of Benzophenone-Type UV Filters from Water Matrices by Advanced Oxidation Processes

**DOI:** 10.3390/molecules27061874

**Published:** 2022-03-14

**Authors:** Belma Imamović, Polonca Trebše, Elma Omeragić, Ervina Bečić, Andrej Pečet, Mirza Dedić

**Affiliations:** 1Department of Drug Analysis, Faculty of Pharmacy, University of Sarajevo, 71000 Sarajevo, Bosnia and Herzegovina; elma.omeragic@ffsa.unsa.ba (E.O.); ervina.becic@ffsa.unsa.ba (E.B.); mirza.dedic@ffsa.unsa.ba (M.D.); 2Faculty of Health Sciences, University of Ljubljana, Zdravstvena pot 5, 1000 Ljubljana, Slovenia; polonca.trebse@zf.uni-lj.si; 3Hospital Pharmacy, University Clinical Center Tuzla, Trnovac bb, 75000 Tuzla, Bosnia and Herzegovina; andrej.pecet@gmail.com

**Keywords:** benzophenones, photodegradation, advanced oxidation processes, photocatalytic degradation

## Abstract

Benzophenone (BP) type UV filters are common environmental contaminants that are posing a growing health concern due to their increasing presence in water. Different studies have evidenced the presence of benzophenones (BP, BP-1, BP-2, BP-3, BP-4, BP-9, HPB) in several environmental matrices, indicating that conventional technologies of water treatment are not able to remove them. It has also been reported that these compounds could be associated with endocrine-disrupting activities, genotoxicity, and reproductive toxicity. This review focuses on the degradation kinetics and mechanisms of benzophenone-type UV filters and their degradation products (DPs) under UV and solar irradiation and in UV-based advanced oxidation processes (AOPs) such as UV/H_2_O_2_, UV/persulfate, and the Fenton process. The effects of various operating parameters, such as UV irradiation including initial concentrations of H_2_O_2_, persulfate, and Fe^2+^, on the degradation of tested benzophenones from aqueous matrices, and conditions that allow higher degradation rates to be achieved are presented. Application of nanoparticles such as TiO_2_, PbO/TiO_2_, and Sb_2_O_3_/TiO_2_ for the photocatalytic degradation of benzophenone-type UV filters was included in this review.

## 1. Introduction

Ultraviolet (UV) light, which comes mainly from the sun, causes damage to materials that are exposed to it. UV light is divided into two subcategories which have wavelengths of 290–320 (UVB) and 320–400 nm (UVA).

Photons of UV light cause breakages of covalent bonds and thus induce different oxidation processes, which are mainly chain-radical oxidation reactions with air oxygen. These processes lead to ageing of different construction materials, coatings, plastics, rubber, etc. [1]. These processes are particularly harmful, however, in biological systems, where they cause damage to skin cells, resulting in accelerated ageing of the skin and the emergence of various diseases, from inflammatory processes to cancer [2,3,4].

To protect against UV light, numerous substances are used that either deflect or absorb UV light in various applications. These compounds are usually called UV filters. Many among them are industrial applications, where the products are exposed to solar radiation such as coating products, plastic products, and cosmetic products. Increasingly, however, they are also used as personal care products (e.g., sunscreen, lipsticks, shampoos, and hair sprays) because of the growing awareness of the harmful exposure to the sun and the consequent increased risk of morbidity for skin cancer. They protect the human body against the harmful effects of sunlight. In addition to inorganic pigments (e.g., TiO_2_), which reflect UV light, there is a large group of organic compounds, which absorb UV light. Because UV light is of a broad spectral range, 290–400 nm (UVA and UVB), no compound can prevent the exposure to the whole spectrum by itself, since their absorption peaks are much narrower [5,6]. Therefore, a combination of several compounds covering the whole UV range is usually applied. Based on the literature survey about the use and effects of old and new formulations, the list of substances permitted by law is regularly updated. The European Union (EU) currently allows 26 organic substances, while others, which are treated as biological agents, are allowed without prescription and regulation in countries around the world, such as Japan and the U.S. [7]. The complete list of UV filters allowed in cosmetic products from Annex VI of the Regulation (EC) No 1223/2009 of the European Parliament and of the Council, updated on 22 September 2021 is available on the EU website [8].

Compounds offering protection from the sun are always used in combination since a single UV filter that could provide a sufficiently high sun protection factor (SPF) does not exist. Trends are going in the direction of increased use of inorganic UV filters, especially in sunscreens for children and creams to protect very sensitive skin. The most used is TiO_2_, which prevents a reduction in SPF, which occurs due to the potential instability of some organic UV filters. However, due to photosensitivity and the potential synergistic effects, international health organizations, e.g., U.S. Food and Drug Agency (FDA, Silver Spring, MD, USA), limit the combinations of different UVA and UVB organic chemical filters [9].

Organic UV filters are usually classified into different categories according to the structure (Table 1) Additionally, they can be divided into two groups, depending on the spectral range covered. The first group consists of so-called UVA filters, which include benzophenone (BP), anthranilates, and dibenzoylmethanes, and in the second one are UVB filters, which include PABA derivatives, salicylate, cinnamates, and camphor derivatives [6].

Besides UV filters, sunscreens may contain other additives such as antioxidants, which are also thought to play role in protecting the skin from the effects of exposure to UV light [10,11,12,13,14].

Compounds that are ranked among the organic UV filters, protecting against sunlight, express characteristics of persistent organic pollutants (POPs). The common characteristic of all these compounds is the presence of aromatic moiety with a side chain, and various degrees of saturation [6]. In general, commercial formulations of sunscreens are comprised of mainly geometrical (*E*)-isomers (trans-), but many of them contain a mixture of both isomers (*E*) and (*Z*)-isomers (cis-) (for example methoxycinnamate). Because of their high lipophilic character (log K_ow_ 4–8) and relative stability against biological decomposition, they were found to also accumulate in the food chain (detection of some UV filters in fish was in the range of 25–1800 ng/g, and in the fat of human milk was in the range of 16–417 ng/g) [6].

### 1.1. Mechanism of UV Absorption

When UV filters are exposed to solar irradiation, they may interact with the electromagnetic rays by absorption or scattering energy. Inorganic particulates, such as TiO_2_ or ZnO, commonly scatter, or reflect rays, but may absorb UV irradiation as well (Figure 1).

These particulates are semiconductors with high band-gap energy between the valence and conduction band in a range of 380–420 nm. As ingredients in sunscreen formulations, the most common are titanium dioxide and zinc oxide.

Organic molecules, on the other hand, absorb UV photons, and electrons in their highest occupied molecular orbital (HOMO) are promoted to their lowest unoccupied molecular orbital (LUMO). The single state obtained may be deactivate through different processes: by simple vibrational relaxation, through fluorescence, through intersystem crossing leading to triplet excited state, or undergoing photochemical reactions.

Photochemical reactions lead also to a change in the physical attributes, such as color, appearance, or to the change of chemical properties, most commonly leading to undesirable reactions and the formation of by-products [1].

One of such reactions is trans-cis isomerization, where the *trans* form has a higher absorption coefficient than the *cis* form, for example in the case of ethylhexyl methoxycinnamate. Another possible photochemical reaction is keto-enol tautomerization, where diketo form absorbs in the UVC range as it occurs in the case of avobenzone. Its enol form exhibits excellent UV absorption at 357 nm, but the diketo form shifts the absorption maximum into the UVC region. For that reason, in such a form it is not an effective UVA or UVB filter. It may react with some other molecules and form photo-adducts [1].

### 1.2. Benzophenone-Type UV Filters

Benzophenone derivatives have the same chemical backbone in common, and their properties and modes of application are the result of differing functional groups. For example, ketoprofen belongs to the benzophenone group of chemicals but is a pharmaceutical compound belonging to the nonsteroidal anti-inflammatory drug (NSAID) class. On the other side, a large group of substituted benzophenones are UV filters and common ingredients of sunscreens and other cosmetic products (Table 2). Some others may be used as photostabilizers in coatings, adhesives, and agrochemicals, etc. [5,15].

Benzophenone-type UV filters share the common structure in which two benzene rings are linked by a carbonyl group (Table 2). Such a conjugated structure enables the absorption of UV radiation efficiently through π-π* and n-π* excitation. Resonance delocalization in the benzophenones is facilitated by the presence of an electron releasing group in *ortho* or/and *para* position, resulting in two λ_max_ at 286 (UVB) and 324 nm (UVA) [1].

The water solubility (at 25 °C) of benzophenones ranges from moderate (30.5 mg/L for BP-6) to high (1.91 × 10^3^ mg/L for 4,4′-dihydroxybenzophenone (4-DHB)). It is assumed that the presence of the methoxy group affects it since the methoxy group increases the octanol/water partition coefficient (log K_ow_) from 4-DHB (log K_ow_ = 2.19) to BP-6 (log K_ow_ = 3.90) as well as the bioconcentration factor (BCF). The acid dissociation constant (pK_a_) values vary from 6.74 to 7.85. Regarding boiling point (about 400 °C) and vapor pressure (from 3.44 × 10^−10^ to 6.62 × 10^−6^ mm Hg), BPs are not volatile chemicals, and for that reason, the loss through volatilization from water is not significant [17].

The UV filters must be relatively stable when exposed to UV radiation. Sunscreen products are used primarily in settings, such as swimming in the sea, swimming pools, and in the snow and in the mountains, where thorough protection is needed. However, several studies showed that these compounds are degraded by light. This occurs mostly through two types of reactions: (a) direct photolytic reactions, and (b) chlorination of aromatic rings or side chains, which is due to the presence of chlorine and a chlorate medium (such as those found in pools, or salty seawater). Information on the fate of these compounds, after they enter the environment, shows the direct release comes mainly from human recreational activities (e.g., swimming and bathing), industrial wastewater discharges and laundry. In addition to washing off directly from the skin and clothes, other routes of transfer to the environment (polishing and washing of cars, textiles) are an important source of contamination [5,18]. Indirect releases from wastewater treatment plant (WWTP) effluents are the main contamination route of BPs, with reported concentrations up to hundreds of μg/L [19]. They have also been detected in various water compartments, like surface and groundwaters and even in drinking water. The occurrence of BPs has been reported in various environmental matrices, such as river water, lakes, groundwater, sediments, suspended particles, and biota [16,17,20].

When these chemicals enter the aquatic environment, they can also cause adverse biological effects on aquatic organisms through toxicity and estrogenic activity. These adverse effects can be the result of the action of the original chemicals, or through their degradation intermediates [21].

In this survey, we focused on stability and photodegradation studies of benzophenone-type UV filters in water compartments as pure substances. Additionally, we have focused on different degradation studies based on advanced oxidation methods like photocatalysis, and combinations of oxidants and UV light. Chlorine-based oxidation processes are not covered in this review.

## 2. Photodegradation of Benzophenone-Type UV Filters in Aquatic Environments

BP-type UV filters are synthesized to protect against UV light. They are used in water-based activities, such as swimming, and for that reason, they are expected to be photostable in aquatic environments. They are lipophilic compounds and therefore tend to bioaccumulate. In the case of BP-3 as per its relatively high log K_ow_ value, i.e., 3.8 (Table 2), the slow biodegradation, tendency to adsorb to suspended solids and sediments, and low volatilization potential from water surfaces are expected. Several studies in the past have dealt with the transformations of UV filters in the aquatic environment, under natural as well as simulated conditions in the presence of sensitizers and confirming their high photostability.

### 2.1. Effect of Natural and Artificial UV Radiation on Benzophenone Stability

One of the first studies, conducted in 1992, has shown that BP-3 degrade only about 4% after 28 days in water [22]. This study was followed by Ricci et al. [23], who studied photostability of selected UVB filters, including BP-3 under UVA irradiation in the presence and absence of TiO_2_. Results of the experiments have confirmed the photostability of BP-3 in the absence of TiO_2_, and the presence of TiO_2_ caused mineralization of UV filters. The process was faster in the presence of surfactants. Rodil [24], later, performed a photostability experiment in which he exposed selected UV filters’ solutions (among several non-benzophenone-type ones also BP-3) to artificially simulated sunlight coupled with a halogen lamp (290 to 800 nm) for a defined period. Light intensity above the water surface was about 350 µmol photons/m^2^. He exposed UV filters in ultrapure water to UV light for different periods (from 5 to 72 h). Three filters, among them BP-3 indicated high stability during the whole irradiation period of 72 h and confirm its photostability.

### 2.2. Main Pathways of Transformation and Kinetics in Different Waters

Vione [25] indicated that the processes which governed BP-3 phototransformation were reactions with hydroxyl radicals (•OH) and therefore the excited triplet states of chromophoric dissolved organic matter (^3^CDOM*). The kinetic parameters, relevant for the photochemical processes involved BP-3 in surface waters (direct photolysis and reaction with •OH, CO_3_^−•^,^1^O_2_, and ^3^CDOM*) were determined by laboratory measurements. The half-life of BP-3 in surface waters was estimated to be several weeks during summer (and 7–9 times longer during winter), and it increased with increasing water depth and increasing dissolved organic carbon (DOC). Authors suggest that BP-3 in surface waters are principally degraded by direct photolysis and reactions with •OH and ^3^CDOM*. Some BP-3 transformation products were identified upon reaction with •OH, among them benzoic acid (produced at a maximum concentration of ~10% of initial BP3) and benzaldehyde (~1% of initial BP3) [25].

In another study, Li et al. [26] investigated the photodegradation of sunscreen agents and their metabolites in pure water, fresh water, and seawater. As model substances, they chose BP-3 and its human metabolite 4-hydroxybenzophenone (4-OH-BP-3). Results show that only anionic forms of both BP-3 and 4-OH-BP3 can undergo direct photodegradation. Indirect photodegradation was a result of reactive species, mainly by DOM making important contributions to the photoinduced transformation of both compounds. In seawater, indirect photodegradation can be especially attributed to ^3^DOM*, while in freshwater, ^3^DOM*, and •OH are responsible for their indirect photodegradation.

Additionally, photodegradation was evaluated for BP-3 and BP-1 as a promising alternative to conventional aerobic bacterial degradation. Photochemical experiments were carried out in a Duran glass UV reactor equipped with a Xenon arc lamp providing a light intensity of 400 W/m^2^. The results have shown in the case of BP-3 that the photodegradation was not efficient. BP-3 remained persistent after 24 h of simulated sunlight irradiation. BP-1 is readily photodegraded and disappears after 24 h under UV radiation. Fungal treatment resulted in the degradation of more than 99% for both sunscreens in less than 24 h. BP-1, on the other hand, has been found as a minor by-product of BP-3 degradation by fungi [27].

Kotnik [28] exposed six benzophenone-type UV filters in laboratory-scale irradiation experiments to a medium pressure UV lamp. The results have shown that photodegradation of benzophenones follows pseudo-first-order kinetics. UV filters were resistant to UV light with a half-life (t_1/2_) between 17 and 99 h. Additionally, natural sunlight exposure experiments (performed in distilled water, lake, and seawater) showed similar photostability as predicted under laboratory conditions and revealed that photosensitizers present in natural waters significantly affect the photolytic behavior of the investigated compounds. Photodecomposition in the lake water was accelerated, while in seawater there were different effects on photodegradation, depending on a compound [21].

### 2.3. Effect of the Presence of BPs on the Degradation of Other Pollutants

The presence of UV filters may also affect the degradation of other pollutants. During the photodegradation of benzotriazole (an anticorrosion agent) and UV filter BP-3 as co-solute, their interactions in aqueous solutions under UV and artificial solar light in the presence/absence of humic acids and metal ions, such as Cu^2+^ and Fe^3+^ have been investigated. Benzotriazole was found to degrade under UV radiation but remained photostable under solar light. Solar irradiation for 50 days resulted in only an 8% decrease in concentration in pure aqueous solution, but up to 31% in the case where humic acids have been added (50 mg/L). One major photoproduct has been identified as 2,4-dimethylanisole, generated by the cleavage of hydroxyl and benzoyl functional groups [29].

### 2.4. Photostability of BPs in Swimming Pool Water

Swimming pool water requires disinfection to prevent swimmers from pathogenic microorganisms, as such reactions of UV filters with disinfectants are unavoidable. Sakkas [30] was the first to report on reactions between UV filters and chlorine in water samples. Negreira [31] studied the stability of various UV filters, among them BP-3, in chlorinated waters and identified corresponding halogenated by-products and assessed their stability in relation to different pHs. These processes are well documented in other reviews. A photostability study of BP-3, BP-4 as well as their chlorinated products (3Cl-BP-3, 5Cl-BP-3, 3,5-diCl-BP-3) in water was studied by Zhuang [32]. The results revealed different stability of each compound in the presence of the UV-A light (photoreactor with 6 UV lamps, 355 nm) after 120 min of exposure. The parent compounds, BP-3 and BP-4, were reduced to less than 5% of their initial concentration within 120 min of irradiation time [32].

### 2.5. Photostability of BPs in the Presence of Organic Dissolve Matters

One of the latest studies, performed by Zhang [33] deals with the degradation of four typical BPs in the UV/nitrite process. The reaction of BPs with •OH and NO_2_• formed hydroxylated and nitrated products. NOM played a negative role in the degradation of BPs by UV/nitrite. This may be due to its light screening and radical scavenging effects. It was also found that the pseudo-first-order rate constants of BPs degradation in the UV/nitrite process increased with increased nitrite concentration. Degradation of BPs was due to the reactions with •OH and •NO_2_ generated by nitrite photolysis. Because of the presence of an electron-withdrawing sulfonate group in the molecule, BP-4 was less reactive to •NO_2_ while •OH played a dominant role in its degradation [33].

Semones et al. [34] investigated the photochemical fate of BP-3 and BP-4. Direct photolysis did not represent an important degradation process, while enhanced photodegradation of both UV filters was observed under simulated solar irradiation conditions in the presence of humic substances, in filtered wastewater effluent and in river water at pH 7. Additional quenching experiments with isopropanol have been performed and confirmed the main degradation pathway is the reaction with the hydroxyl radicals (•OH). The 24-h averaged half-lives near the surface were calculated to 3.0 for oxybenzone (BP-3), and 4.0 days for sulisobenzone (BP-4), respectively. When extrapolated to an environmentally representative water column, this same 24-h averaged half-lives increased to 2.4 and 3.5 years, respectively [34].

All studies confirmed the high persistence of benzophenone-type UV filters towards UVA as well as UVB light. For that reason, finding a proper method for their removal is of high interest and importance. Several advanced oxidation processes have been examined for this purpose and are presented in the following chapter.

## 3. Degradation of Benzophenone-Type UV Filters Based on Advanced Oxidation Method

Benzophenone-type UV filters are poorly biodegradable compounds [35] and many of them (e.g., BP-3, BP-4, and BP-8) have been detected in the aquatic environment due to their higher solubility in water and relatively low log K_OW_ [36,37,38,39].

Conventional water treatment processes, which include coagulation, sedimentation, filtration, and chlorination, are unable to remove this type of micropollutant from drinking water, wastewater, and water used for irrigation [40]. Considering the potential hazards of the presence of this type of micropollutants in water, and being a source of potential hazard to humans and the environment, various treatments have been developed for their removal. Advanced oxidation processes (AOP) are often used today in water treatment with the aim of removing organic micropollutants from water [41]. Ultraviolet (UVC) irradiation, UV/H_2_O_2_, UV/persulfate, photo-Fenton, and photocatalysis are the most common chemical technology processes used to remove organic UV filters from water (Figure 2), due to their high oxidation power [42,43,44,45]. UV/H_2_O_2_ process, UV/Fenton process, UV/persulfate process, as well as photocatalysis as processes for the removal of benzophenone-type UV filters are discussed in detail.

### 3.1. Degradation of Benzophenone Type UV Filters: UV/H_2_O_2_ Process

The UV/H_2_O_2_ process is one of the most used AOPs (Figure 2A). UV photolysis of H_2_O_2_ produces two •OH radicals in the system. These radicals can react with each other to form new H_2_O_2_ molecules or react with other organic substances in the system. UV radiation plays a major role in the formation of these radicals. An important fact to note is that H_2_O_2_ is not degraded by UV radiation of wavelengths below 254 nm, so UVC radiation must be used. The photolysis of H_2_O_2_ is also influenced by the transparency of the test solution, as well as the optical properties of the UV reactor. The highly reactive hydroxyl radical formed during the photolysis of H_2_O_2_ can oxidize organic substances to small molecules that are biodegradable [42,46]. Benzophenone-3 (BP-3) can be very easily degraded under the influence of UV/H_2_O_2_ processes in an aqueous solution. It has been shown that photolysis in the presence of H_2_O_2_, after 8 h under UV irradiation at 254 nm, causes BP-3 degradation. The reaction depends on the initial BP-3 concentration and pH. At higher initial concentrations of BP-3, the percentage of degradation decreased, and the optimal pH for this degradation was 6.0. When the concentration of BP-3 was 0.01 mM, the rate constants of the apparent first-order reaction and the second-order reaction between BP-3 and •OH were 1.26 × 10^−3^ s^−1^ and 2.97 × 10^10^ M^−1^ s^−1^, respectively. Several intermediates were identified by GC-MS and the primary reaction pathway between BP-3 and •OH was proposed [47]. Similar studies were conducted on benzophenone-4 (BP-4), benzophenone-9 (BP-9) [48], benzophenone (BP), and 4,4-dihydroxy-benzophenone (HBP), with the aim to remove this type of UV filter from water by UV/H_2_O_2_ process [49].

As with BP-3 [41], aqueous solutions of BP-4 and BP-9 were exposed to the UV/H_2_O_2_ process. Both benzophenones were tested at the same initial concentration, H_2_O_2_ concentration and a certain intensity of UV irradiation. In addition, a control experiment was done, with BP-4 and BP-9 treated with UV light only. UV irradiation alone did not affect the degradation of the tested compounds, which indicated that ozonation alone is not enough to remove this type of pollutant from water. The complete transformation of BP-4 and BP-9 was performed in the presence of UV/H_2_O_2_ systems, within 16 min, indicating high oxidative reactivity between the tested benzophenones and •OH radicals. The concentration changes follow first-order kinetics reaction, with BP-4 degradation occurring faster than BP-9.

Using the LC-ESI-MS/MS technique, 12 BP-4 intermediates and 17 BP-9 intermediates were identified. The toxicity of the produced intermediates was evaluated using a bioluminescent assay using the bacterium *Vibrio fischeri*, where the changes in the luminescence of the bacterium during exposure to toxic substances was monitored. Intermediates of BP-4 and BP-9 showed greater toxicity than parent compounds. Using the ECOSAR program (EPA’s ecological structure–activity relationships), the acute toxicity of BP-4 and BP-9 and their intermediates were predicted and showed higher toxicity of intermediates to living organisms (fish, algae) than BP-4 and BP-9 [48]. This indicates that special attention should be paid to the formation of benzophenone intermediates and that a risk assessment should be done in future research.

Unlike the tests performed on BP-4 and BP-9, tests performed on BP and HBP [49] in the presence of UV/H_2_O_2_ differed in methodology. The response surface methodology (RMS) and central composite design (CCD) were applied in the study to determine the effects of initial concentration, H_2_O_2_ concentration, and intensity UV irradiation on the degradation process, for BP and HBP, respectively. In both tested UV filters, the reaction under UV/H_2_O_2_ process conditions followed first-order reaction. BP degraded much faster in the UV/H_2_O_2_ process compared to HBP, but the initial concentration was an important factor controlling degradation (with a negative effect), after which the greatest influence on the course of degradation was H_2_O_2_ concentration and UV irradiation intensity (with positive effects) [49]. The prediction made using the RMS model correlated with the experimental data. This indicates that the application of the RMS model could play an important role in the optimization of experimental conditions, in this type of research [50]. Kinetics and degradation pathways, as well as the formation of intermediates, depend on the conditions under which the UV/H_2_O_2_ process takes place. In all examples of benzophenone-type UV filter research, a certain number of intermediates have emerged that may have potential toxic effects on living organisms. The effect of different conditions on the degradation process of BPs, using the UV/H_2_O_2_ system is shown in Figure 3.

### 3.2. The Degradation of Benzophenone Type UV Filters: UV/Fenton Process

One of the most commonly used advanced oxidation processes (AOPs) is the Fenton process, which is based on the oxidation of Fenton reagent (an oxidative mixture of H_2_O_2_ and Fe^2+^ salts as a catalyst) (Figure 2B). During the Fenton process, hydroxyl radicals are formed from the reaction of Fenton reagents (iron salt and H_2_O_2_) in an acidic medium (pH about 3). At a higher pH, ferrous ions (Fe^2+^) are converted to ferric ions (Fe^3+^) which then act by binding to hydroxyl ions and precipitating from the system as iron hydroxide. The presence of a higher concentration of H_2_O_2_ in the system is responsible for the precipitation of iron hydroxide, and therefore the Fenton process must take place at a lower pH. Fenton’s reagent is a strong oxidizing agent based on the binding of OH^−^ radicals with organic compounds. The working principle of the Fenton process is shown by the following equations:
H_2_O_2_ + Fe^2+^ → Fe^3+^ + OH^−^ + •HO(1)
Fe^3+^ + H_2_O_2_ → Fe^2+^ + •HOO + H^+^(2)

Iron (Fe) is oxidized with hydrogen peroxide to Fe^3+^, forming a hydroxyl radical and a hydroxide ion in the process. Fe^3+^ is reduced back to Fe^2+^, forming a hydroperoxyl radical and a proton. The decomposition of organic molecules takes place according to a very complex mechanism that includes oxidation by hydroxyl radicals, direct oxidation with hydrogen peroxide and oxidation with other radicals, and mutual reactions between organic radicals.

Unlike the ordinary Fenton process (Fe^2+^/H_2_O_2_), the addition of UV light to the Fenton process can produce more hydroxyl radicals through the photoreduction of ferric ions into ferrous ions and the direct UV/H_2_O_2_ reaction. Iron ions produced by the photo-Fenton process can further react with H_2_O_2_ to form more radicals in the system [42]. This process is widely used in the removal of organic pollutants from water, including benzophenone-type UV filters. BP-3 can be removed from an aqueous solution very successfully by UV/Fenton process, considering the optimization of the conditions of the process itself. The study conducted on BP-3 [51] to remove this pollutant from the aqueous matrix was carried out at precisely determined initial concentrations of Fe^2+^, H_2_O_2_, and UV radiation intensity. To determine the optimal conditions, a face cantered, central composite design was carried out. Results indicated that the BP-3 degradation rate and reaction kinetics depends on the concentrations of Fe^2+^, H_2_O_2_ and the intensity of UV radiation. How an increase in the concentration of BP-3 affects the course of the reaction was also evaluated and it was found that an increase in the concentration of BP-3 increases the rate of degradation, and the reaction is a pseudo-first-order kinetic reaction. BP-3 was exposed to UV irradiation in the range of 300–800 nm, because the process of •OH radical formation is only possible above 254 nm [46], and the pH of the solution was close to 3 (2.85–3.1). Under optimized conditions, the degradation of BP-3 was complete in 60 min, as opposed to the traditional Fenton process (Fe^2+^/H_2_O_2_). BP-3 degraded to CO_2_ and H_2_O, and the mineralization process was quantified using a TOC analyzer, a device for determining dissolved organic carbon (DOC). The BP-3 biodegradation study was conducted via chemical oxygen demand (COD) and biochemical oxygen demand (BOD5). BOD_5_/COD ratio analysis showed that after 300 min of treatment 61% of the substrate was demineralized and solution biodegradability increased gradually [51]. This research showed that the UV/Fenton is an excellent process for the elimination of BP-3 from aqueous matrices. From this class of UV filters, the stability of BP-1 and BP-2 was investigated under the influence of the UV/Fenton process [52] in conditions very similar to BP-3 [51]. Aqueous samples containing BP-1 and BP-2 were exposed to Fenton reagent and UV irradiation. The samples were analyzed at different time intervals, after quenching remnant (neutralizing remaining?) H_2_O_2_. The resulting degradation products were analyzed using a linked LC-MS system, and the NIST database was used to identify degradation products of BP-1 and BP-2 formed during the UV/Fenton process. Dissolved organic carbon (DOC) present in the tested water samples was determined using a TOC analyzer.

Chemical oxygen demand (COD) and biochemical oxygen demand (BOD_5_) were calculated according to the methodology described in the Standard Methods for Testing Water and Wastewater, 2012, according to methods 5220 D and 5210 D [52,53].

The experiment showed that with the help of the UV/Fenton process it is possible to remove benzophenone type UV filters: BP-1 and BP-2, both individually and in a mixture, from aqueous matrices. In addition, demineralization and biodegradation of the sample can then be performed. The BOD_5_/COD ratio was analyzed for BP-1, BP-2, and their mixture. After 300 min of exposure, the achieved extent of mineralization was 64% for individual UV filters, while mineralization in the mixture occurred more slowly.

An explanation for this could be higher competition between the compounds present in the solution for the oxidizing radicals.

During the UV/Fenton process, the degradation products are formed under the influence of •OH radicals by the hydroxylation process, which is the first step in the oxidation of BP-1 and BP-2. The resulting degradation products of BP-1 were: 2,2,4-trihydroxybenzophenone, 2,2,4,4-tetrahydroxybenzophenone, benzaldehyde, resorcinol, 4-methylphenol, phenol, 2-methylphenol, 2-hydroxybenzaldehyde, acetic acid and formic acid. Degradation of BP-2 resulted in benzaldehyde, resorcinol, 1,2,3-benzenetriol, 4-methylphenol, phenol, 2-methylphenol, 2-hydroxybenzaldehyde, acetic acid and formic acid.

The initial concentrations of H_2_O_2_ and Fe^2+^ ions play a very important role throughout the whole process. The interaction of these factors has an initially positive influence on the degradation of the tested substances; however, the excess of H_2_O_2_ and Fe^2+^ ions may generate a process that leads to the removal of •OH radicals [53]. The UV/Fenton process is a process that can successfully eliminate BP-1 and BP-2, but great care must be taken to ensure appropriate concentrations of H_2_O_2_ and Fe^2+^, which can have a negative effect on the elimination of the tested UV filters from water matrices, as well as the toxic effects of BP-1 and BP-2 degradation products. The effect of different conditions on the degradation process of BPs, using the UV/Fenton system is shown in Figure 4.

### 3.3. The Degradation of Benzophenone Type UV Filters: UV/Persulfate Process

AOPs based on sulfate radicals have recently attracted attention due to certain advantages, such as high redox potential, longer radical lifetime, and lower sensitivity to the scavenging effect [54], which is present in UV/H_2_O_2_ and UV/Fenton process. Sulfate radicals are generated through the activation of peroxymonosulfate (PMS) or persulfate (PS) ions (Figure 2C). Activation methods include heat, transition metals, ultrasonic radiation, and UV irradiation. Among all activation processes, UV irradiation is of great importance [55]. Sulfate radicals are formed due to the breakage of the peroxide (–O–O–) bond in PS or PMS. Similarly, hydroxyl radicals are generated due to the decomposition of hydrogen peroxide (Equations (3)–(7)).
S_2_O_8_^−2^ + UV/heat/ultrasound → 2 •SO_4_^−^(3)
S_2_O_8_^−2^ + Fe^+2^ → Fe^+3^ + •SO_4_^−^ + SO_4_^−2^(4)
S_2_O_8_^−2^ → 2•SO_4_^−^(5)
•SO_4_^−^ + H_2_O → •OH + HSO_4_^−^•(6)
•SO_4_^−^ + Fe^+2^ → Fe^+3^ + SO_4_^−^ + SO_4_^−2^(7)

The energy of the peroxide (–O–O–) bond in PMS/PS and hydrogen peroxides is 140 and 213.3 kJ/mol, respectively. This means that the formation of sulfate radicals requires less energy, compared to •OH radicals in the presence of UV irradiation [55]. Decomposition of PS or PMS produces sulfate and •OH radicals in the system. One molecule of PS produces two, while one molecule of PMS produces one sulfate radical [55]. UV/PS is effective for decomposition as well as mineralization of organic pollutants in milder conditions. This process has been investigated for the removal of various organic pollutants [56,57]. In UV/PS system, pH is an important parameter because sulfate radicals are dominant species at a lower pH [58]. Degradation of BP-3 in an aqueous solution was examined in the presence of UV irradiation and PS [59].

The influence of UV and visible light irradiation, transition metal ions, and heat on the PS system activation at different pH values was investigated. The influence of heat was examined at temperatures ranging between 25 and 40 °C, and the activation tests with transition metal ions Fe^2+^, Cu^2+^, and Co^2+^ were conducted at different concentrations. The results showed that UV irradiation, heat, and transition metals used to activate PS greatly affect the elimination of BP-3 from aqueous solution. The heat has an exceptional effect on the activation of the PS system, regarding the formation of SO_4_^−^ ions. After 3 h, at a temperature of 25 °C, only 11% of BP-3 was removed, while at a temperature of 40 °C BP-3 was eliminated from the aqueous solution. UV radiation leads to breaking O–O bonds in PS and to increased production of •SO_4_^−^ ions. BP-3 degradation is significantly increased due to light exposure. The wavelength to which the solution was exposed plays an important role in PS activation and BP-3 degradation. Short wavelength irradiation provided more energy for PS activation in comparison to long wavelength. PS can be activated by transition metals [60], for this purpose Fe^2+^, Cu^2+^, and Co^2+^ were used, and the results showed different levels of PS activation. The positive effect of these cations on the BP-3 degradation was in the following order: Fe^2+^ > Co^2+^ > Cu^2+^. It was observed that the BP-3 degradation by Fe^2+^/PS was faster than Cu^2+^/PS, especially at higher concentrations of Fe^2+^, which leads to the conclusion that the initial conversion of Fe^2+^ plays an important role in the activation of PS. However, care must be taken, because although Fe^2+^ can react with PS and generate a large amount of •SO_4_^−^ ions, excess Fe^2+^ can react with •SO_4_^−^ ions, and thus inhibit the reaction itself [59,61]. Unlike Fe^+2^/PS, the generation of •SO_4_^−^ ions by Cu^2+^/PS is slower and requires more energy. When the metal ion concentration was 0.01 mM, the BP-3 degradation efficiency was highest for Co^2+^, but with increasing concentration of Co^2+^ cation, the BP-3 degradation efficiency changed. Co^2+^should be used with extreme caution as it poses a health risk at higher concentrations [59]. The BP-3 degradation was also examined in the pH ranges of between 3.0–12.0. Except for pH 12.0, degradation decreased with increasing pH. This can be explained by the concentration and type of radicals, the formation of which depends on the pH [59,61]. At pH < 7.0, the •SO_4_^−^ ion is a radical that is dominant in the system, and its generation is favored by the effect of acid catalysis. The resulting •SO_4_^−^ ion at acidic pH can react with BP-3 and increase the degradation effect.

The highest degradation efficiency was observed at a pH of 12.0, which can be explained by the fact that the dominant •OH radical is in the PS system at this pH value. At a higher pH, the •SO_4_^−^ radical, formed from PS by base activation, can be converted to •OH radicals. The •OH radical has a slightly higher redox potential than •SO_4_^−^ in the base medium, resulting in a higher BP-3 degradation rate [59]. The initial concentration of PS is an important step in the degradation efficiency of BP-3. The effect of initial PS concentrations on BP-3 degradation was examined. The tests were conducted at the BP-3: PS molar ratio of 1:100; 1:250; 1:500, and 1:1000. Enhanced oxidation occurred as the PS concentration increased, which can be explained by the increased concentration of •SO_4_^−^ radicals.

The largest enhance occurred at a ratio of 1:500, but with an increase in PS concentration to 1:1000, no positive effect of BP-3 removal was observed with increasing PS dose, i.e., the removal efficiency was not proportional to the increased PS concentration [59]. Seven by-products of the BP-3 degradation via UV/PS system were identified and degradation pathways have been proposed. Hydroxylation, demethylation, and direct oxidation are the basic processes that were involved. The toxicity of the resulting degradation products was evaluated using the ECOSCAR program developed by USEPA. It has been found that the degradation products of BP-3 formed during the UV/PS process are less toxic than BP-3 itself, which puts this method for removing this type of pollutants from water at the top of the list of methods for their elimination [59].

The effect of bromide on the BP-4 degradation via UV/PS/Fe^2+^ system was investigated together with ampicillin and benzene derivatives, as these are very common pollutants that are resistant to the conventional wastewater treatment processes [62,63]. BP-4 easily enter the environment from swimming pools and tests have shown that it can have a xenoestrogenic effect on aquatic organisms [64]. The influence of PS, Fe^2+^ and Br^−^ on the BP-4 degradation rate in the aqueous matrix were evaluated. Increasing the concentration of Fe^2+^ ions in the presence of a constant concentration of PS and Br^−^ did not enhance the degradation of BP-4 after a time of 200 min. On the other hand, the increase in PS concentration significantly enhanced the BP-4 degradation rate, which could be attributed to the generation of more •SO_4_^−^ radicals. The influence of Br^−^ on the BP-4 degradation rate was not of great importance, except that the BP-4 degradation rate in the presence of bromide ions slowed down a bit and followed the kinetics of the first order, in the presence and absence of Br^−^, respectively. In addition, the influence of the pH on the degradation rate was evaluated, and the results showed that the pH adjustment did not have any significant influence on the BP-4 degradation rate. From this, we can conclude that the UV/PS/Fe^2+^ system is effective for the removal of BP-4 from wastewater [62]. The effect of different conditions placed on the degradation process of BPs, using the UV/persulfate system is shown in Figure 5.

### 3.4. Application of Nanoparticles for the Photocatalytic Degradation of Benzophenone-Type UV Filters

Although UV filters contain chromophore groups that can absorb light at different wavelengths in the UVA in the UVB range, they represent very stable molecules.

Literature data show photolysis using natural or artificial UV light does not take place and the degradation processes are very slow as was described previously in Section 2.

Heterogeneous photocatalysis is one of the successful approaches for the elimination of organic compounds from environmental samples. Usually, catalyst nanoparticles are immobilized into monolith structures and due to this a post-filtration step is not necessary as it should be involved in slurry suspensions [65].

Several semiconductors have been used as photocatalysts and among them, TiO_2_-P_25_ seems to be the most photoactive with a bandgap of 3.2 eV under UV irradiation [65,66].

#### 3.4.1. Fundamentals and Mechanism of TiO_2_ Photocatalysis

According to Chong [67], the mechanism of TiO_2_ photocatalysis could be described by the series of chain oxidative/reductive reactions beginning with the photoexcitation reaction after the illumination of the TiO_2_ surface with the photon energy (*hv*) greater than or equal to its bandgap energy, creating a photogeneration of holes (in the semiconductor valence band) and electron pairs (conduction band) (Equation (8)) [68]:
TiO_2_ + *hv* → TiO_2_^−^ + OH• (or TiO_2_^+^)(8)

Energized holes and electrons may recombine and dissipate the absorbed energy as heat (Equation (9))
TiO_2_^−^ + OH• + H^+^ → TiO_2_ + H_2_O (recombination)(9)

They may also be available for use in the redox reactions (Equations (10)–(12)).
TiO_2_^−^ + O_2_^+^ → TiO_2_ + HO_2_(10)
TiO_2_^−^ + H_2_O_2_ + H^+^ → TiO_2_ + H_2_O + OH(11)
TiO_2_^−^ + 2H^+^ → TiO_2_ + H_2_(12)

For heterogeneous photocatalysis, if the irradiation time is extended, the liquid phase organic compounds are degraded to their corresponding intermediates and further mineralized to CO_2_ and H_2_O (Equation (13)) [69].
OH• + O_2_ + C_x_O_y_H_(2x−2y+2)_ → x CO_2_ + (x − y + 1) H_2_O(13)

#### 3.4.2. Photocatalytic Degradation of Benzophenone-Type UV Filters

Several studies have reported the application of TiO_2_ as a catalyst in the degradation process of benzophenones (as an active ingredient in sunscreens). In most of these studies the effects of catalyst surface area, catalyst amount, pollutants (benzophenones) concentration, pH, contact time, and the presence of hydrogen peroxide on the degradation of the pollutant were examined (Figure 2D).

##### Effect of Catalyst Amount on Photocatalytic Degradation

Celeiro et al., (2019) [70] assessed the performance of different photodegradation strategies to simultaneously remove twenty-one multiclass organic filters from water, among them several of those belonging to benzophenone-type UV filters. They compared direct photolysis (UVA and UVC), photocatalysis UVA/TiO_2_, and UVC/H_2_O_2_. It was demonstrated that UVC (germicidal light) is generally the most efficient method with degradation yields higher than 90% after 60 min of exposure in ultrapure water. The degradation rate depends on the type of water, showing lower degradation in the case of the presence of organic matter for example.

Moradi [71] used TiO_2_ nanoparticles, synthesized using thesol-gel method, coated on quartz tubes and applied in the photocatalytic experiment for degradation of BP-3. Details about photocatalytic experiments are presented in Table 3.

**Table 3 molecules-27-01874-t003:** Overview of the photocatalytic reactions used s in degradation reaction of BPs.

Photocatalyst	Pollutant	Optimal Conditions	Degradation Rate	Reference
TiO_2_ nanoparticles coated quartz tubes	BP-3	pH 10, BP-3 concentration 1 mg/L, 225 cm^2^ of catalyst surface area, 15 min UVC irradiation	98%	[71]
TiO_2_ nano-layer on quartz wool (TiO_2_-qw)	BP-3, BP-4	pH 7, 8, initial concentrations 5 mg/L, deionized and tap water, catalyst quartz wool, UVC irradiation, 4 h of treatment	>90% in deionized water 70% in tap water	[72]
TiO_2_ (Degussa P-25)	BP-3	pH 9.0, BP-3 concentration 1 mg/L, TiO_2_ concentration of 1.184 g/L, and H_2_O_2_ concentration of 128.069 mg/L, 30 min UVC irradiation	91.66%	[72]
TiO_2_ nanowires (TiO_2_NWs)	BP-4	pH 5, BP-4 concentration of 20 μM, catalyst concentration 1.2 g/L, 180 min UV irradiation (400–360 nm)	90%	[73]
Cellulose acetate monolithic structures coated with thin films of commercial Fe_2_O_3_and TiO_2_ (P25, PC105, and PC500) nanoparticles with which the photoreactor tube is coated	Ensulizole (PBSA), BP-4 and BP-3	pH 7, concentration of pollutants: 0.042 µM (PBSA), 0.042 µM (BP-4), 0.051 µM (BP-3), H_2_O_2_ of 0.59 mM, 30 min UVA	44% (PBSA), 90% (BP-4), and 91% (BP-3)	[74]
PbO/TiO_2_-2:1	BP-3	pH 7, BP-3 concentration 20 µM, PbO/TiO_2_-2:1 was 0.75 g/L, 120 min UVC irradiation	86.6%	[73]
Sb_2_O_3_/TiO_2_-2:1	BP-3	pH 9, BP-3 concentration 20 µM, Sb_2_O_3_/TiO_2_-2:1 concentration 0.25 g/L, 120 min UVC irradiation	80.3%	[73]

Saracino [72] evaluated the possibility of producing large batches of photocatalyst nano-layers, immobilized on a high-surface-area soiled substrate and their application for the removal of various emerging organic contaminants (among them BP-3 and BP-4). Increasing the amount of catalyst, in general, leads to an increased efficacy rate of the degradation of pollutants. Soto–Vázquez [73] have found that as the catalyst loading increases, higher degradation was obtained reaching approximately 60% when 1.2 g/L was used [73]. This could be explained by two processes. Firstly, increasing the concentration of a catalyst leads to an increased number of active sites, and secondly, when TiO_2_ concentration increases, more photons can be absorbed by a catalyst, and consequently, a larger amount of HO• and other reactive species could be generated which can contribute to the BPs degradation process.

However, if the catalyst concentration exceeds ~1.20 g/L, due to aggregation of TiO_2_ particles there is decreasing the degradation of the pollutant as a result of the reduction in the light penetration and the light scattering [74].

The same effect on degradation rate is obtained by increasing the catalyst surface area. According to Moradi [71], when the surface area of the catalyst increased from 45 to 225 cm^2^, the degradation efficiency increased from 35.7% to 98%. The authors explained this effect by the fact that increasing the catalyst surface area leads to an increase in the quantity of TiO_2_, the number of active sites, and more UV photons can be absorbed by the surface of the catalyst [74].

##### Effect of BPs Initial Concetration on Photocatalytic Degradation

The photocatalytic activity was, generally, enhanced at lower initial concentrations of BP-4 [73] and BP-3 [74]. Moradi [71] demonstrated that by increasing the initial concentration of BP-3 from 1 to 5 mg/L, the degradation efficiency decreases from 98% to 47%. This can be a consequence of catalyst-surface occupation by a higher concentration of pollutants leading to a lack of active sites on the surface of the catalyst. It could also be the case that with increasing the concentration of the solution, UV photons cannot penetrate the solution and its path length becomes shorter, [74] or thirdly it may be associated with the photoproducts formed in the process competing for the active sites at the catalyst [73].

##### Effect of pH on Photocatalytic Degradation

Moradi [71] found that with the increase in pH from 3 to 10 photodegradation efficacy of the BP-3 increased from 17% to 82%. The authors explained this may be caused by generating more •OH ions at the higher pH. These ions are generated on the surface of TiO_2_ and easily changed into •OH in alkaline solutions, and, as in photocatalytic degradation, the removal efficiency is highly affected by the catalyst’s surface charge and substrate’s ionic form so the molecular form of BP-3 in acidic condition (pKa of BP-3 is 8.06) and deproteinated phenolic group under alkaline condition prevents BP-3 to be adsorbed on the catalyst’s surface [74]. Similar to the previous findings Zúñiga–Benítez [74] also found that increasing the pH leads to an increase in the percentage of removal of the BP-3. Additionally, these authors explained the observed effect by effect of pH is related to the surface charge of the catalyst and its relation to the ionic form of the substrate which can lead to a modification of the overall removal rate. The point of zero charge (pzc) of TiO_2_ Degussa P-25 has been reported in the range between 5.7–6.5 and at a pH lower than the pzc then the catalyst surface will be positively polarized, while at a pH higher than the pzc the TiO_2_ charge is negative [74].

This implies that cationic electron acceptors will be favored at pH > pzc conditions, while anionic electron donors will be favored at pH < pzc conditions. Accordingly, it can be expected that the charge of the catalyst surface at a high pH will lead to BP-3 removal.

##### Effect of H_2_O_2_ Concentration on Photocatalytic Degradation

It was also noted that the extent of BP-3 degradation was increased by increasing the H_2_O_2_ concentration, however, this effect is limited at a concentration of ~128 mg/L and a detraction in substrate degradation is observed from that point. This effect (lower concentration (<128 mg/L), improvement in BP-3 removal) is because H_2_O_2_ can be considered a better electron acceptor than oxygen which can promote the generation of •OH free radicals.

H_2_O_2_ also can react with the superoxide anion radical •O^2−^. On the other hand, at higher H_2_O_2_ concentrations (>128 mg/L), the excess H_2_O_2_ scavenges the •OH free radicals generated by photocatalytic reactions and form a much weaker oxidant, hydroperoxyl radical. In addition, •HO_2_ also can react with the remaining •OH to form oxygen and water, resulting in a reduction of the number of available hydroxyl radicals [74].

##### Degradation Kinetics

The degradation of BP-4 under optimal experimental conditions follows the reaction of pseudo-first-order with the kinetic constant of 1.29 × 10^−2^/min, and an R^2^ > 0.99.

According to Zúñiga–Benítez [74] analysis, the kinetics of degradation of BP-3 in the concentration range 0.5–2.0 mg/L are first order and at higher initial concentrations a deviation from first-order kinetics occurs. Since BP-3 concentration in natural waters has been reported in levels around ng/L, it can be assumed that the pollutants photocatalytic degradation in environmental water bodies, follows the first-order kinetics [72]. Table 3 shows an overview of the photocatalytic reactions used in the degradation reaction of BPs with the stress on optimal conditions of the reaction and photodegradation efficacy obtained.

##### Identification of Intermediaries and Toxicity Studies

By the analysis of the content of DOC, it can be noted that total elimination of BP-3 can be reached after 60 min, whereas about 67% of DOC is removed after 300 min. That photocatalysis with TiO_2_ leads to BP-3 transformation into CO_2_ and H_2_O. Additionally, evaluation of biodegradability confirms that TiO_2_ photocatalytic degradation enhances the biodegradability of the solution significantly, and an evaluation of solution toxicity (by measuring the EC_50_ on *Vibrio fisheri*) confirms the reduction in the toxicity of the original solution (around 83%). All these findings suggest that the TiO_2_ photocatalytic system can oxidize and mineralize BP-3 and significantly reduce toxicity while increasing biodegradability [74].

A study conducted by Zúñiga–Benítez [51] identified ten BP-3 by-products: 2,4-dihydroxybenzophenone; 2,2′-dihydroxy-4-methoxybenzophenone; benzaldehyde; 1,3-dihydroxybenzene; 4-methylphenol; benzoic acid; 2-methylphenol; 2-hydroxybenzaldehyde; 1-methyl-2-(phenylmethoxy)-benzene; and benzyl alcohol.

Two main metabolites of BP-3 are 2,4-dihydroxybenzophenone and 2,2′-dihydroxy-4-methoxybenzophenone. The degradation pathway of the BP-3 probably begins with the hydroxyl radicals attack to ortho, meta, and para positions on two benzenic rings in its molecular structure. As a consequence, six by-products (benzaldehyde; 1,3-dihydroxybenzene; 4-methylphenol; benzoic acid; 2-methylphenol; 2-hydroxybenzaldehyde;) are generated. Due to an esterification reaction between benzoic acid and 2-methylphenol, 2-methyphenyl benzoate can be generated. Methyl-2-(phenylmethoxy)-benzene can be attacked by radicals and generates benzyl alcohol. Finally, all these identified phenolic substances are most likely oxidized to different aliphatic compounds and finally to CO_2_ and water [74].

Studies have investigated composite photocatalysts of TiO_2_ and the effects of pH, catalysts loading effect on the performance of synthesized catalysts.

It was found that increasing pH of BP-3 solution leads to the increase of the degradation rate and better degradation was observed at alkaline conditions (pH 7–9) for PbO/TiO_2_-1:1, PbO/TiO_2_-1:2, and Sb_2_O_3_/TiO_2_-2:1 [75].

Under alkaline conditions, the density of OH^−^ ions can be higher, which can promote the generation of OH• by the catalysts which then further drives degradation of BP-3. Among significant parameters for evaluating the performance of synthesized catalysts belongs the catalysts’ dosage loading. Many studies pointed out that to achieve the best degradation rate, the catalysts dosage loading should adjust to a proper concentration, which is dependent on pollutant concentration, pH of the solution, and UV irradiation.

The photocatalytic activity of PbO/TiO_2_ (2:1) increases at lower initial concentrations of BP-3 and with the minimum concentration (20 µM) the BP-3 degradation rate reaches the maximum and by increasing the initial concentration, the degradation rate decreases progressively. Based on the MS fragmentation pentamethyl- and 5-hydroxy-7-methoxy-2-methyl-3-phenyl-4-chrome, none was confirmed as a by-product of the photocatalytic degradation [75].

##### Applicability of the Photocatalyst in BPs Removal from Water

There is a paucity of studies that focus on the removal of UV filters from swimming pool waters, using photocatalysts. Celeiro et al. [76] successfully applied photocatalytic degradation of three UV filters, among them BP-3 and BP-4, in a range of small concentrations (μ/L) in synthetic and real swimming pool water. UVA heterogeneous photocatalysis experiments using commercial TiO_2_ and Fe_2_O_3_ nanoparticles, with the addition of hydrogen peroxide, have been performed. Photocatalysis using monolith structures, coated with TiO_2_-P-25 were shown to be the best in terms of removal efficiency-(>90%) for benzophenones after 30 min. The addition of H_2_O_2_ substantially improved the photocatalytic rate with the complete removal of BP-3 and BP-4 in less than 6 min [76]. The effect of different conditions on the degradation process of BPs, using the UV/TiO_2_ photocatalytic system is shown in Figure 6.

## 4. Conclusions and Future Perspectives

The use of BPs type as UV filters have increased as a consequence of public awareness of the effects of harmful exposure to the sun and the increased risk of skin cancer. This has led to elevated concentrations of these pollutants, not only in the natural environments but also in wastewater treatment plants. Finding a proper method for wastewater treatment is very challenging, especially when other chemicals which may be used for their removal represent additional contaminants in the environment.

AOPs are frequently used in water treatment to remove organic micropollutants from water. UV irradiation, UV/H_2_O_2_, UV/persulfate, photo-Fenton, and UV/photocatalysis are the most common processes used to remove benzophenones from water.

In UV/H_2_O_2_ systems, the kinetics and degradation pathways, as well as the formation of the intermediates, depend on the conditions under which the UV/H_2_O_2_ process takes place. This process uses UV radiation above 254 nm and the sunlight can also be used to provide solar energy. A major disadvantage of this process is the degradation products themselves, which may have toxic effects on living organisms.

In the Fenton process, the initial concentrations of H_2_O_2_ and Fe^+2^ ions play a very important role throughout the process. Initially, these factors have a positive effect on the degradation of the tested BPs, however, an excess of H_2_O_2_ and Fe^+2^ ions can trigger a process that leads to the removal of •OH radicals and thus lead to inhibition of the reaction. The UV/Fenton process requires UVA irradiation which can be provided with sunlight. In conclusion, the UV/Fenton process can successfully eliminate BP-type UV filters, but to optimize the method care should be taken with concentrations of H_2_O_2_ and Fe^+2^ that may adversely affect the elimination of tested UV filters from matrix water. Although it has its benefits, the degradation rate of BP that are achieved with the UV/Fenton system is lower in comparison with other degradation processes analyzed within this work. To date, there are no data on BP’s intermediates or their toxicological evaluation, making a full assessment of their environmental usefulness difficult.

AOPs based on sulfate radicals have recently attracted attention due to certain advantages such as high redox potential, longer radical life, and lower sensitivity to the cleaning effect found in the UV/H_2_O_2,_ and UV/Fenton process. Sulfate radicals are formed by the activation of PMS or PS ions. Activation methods include heat, transition metals, ultrasonic radiation, and UV radiation. Among all the activation processes, UV radiation, heat and the type of transition metals used is of great importance. All the AOPs processes can be successfully used to remove benzophenone UV filters from water, but care must be taken about the conditions under which a particular process takes place and the concentration of reagents applied in certain AOPs processes.

Photocatalysis seems a promising method for the degradation of various pollutants, including benzophenone-type UV filters. The results of available studies have shown TiO_2_ photocatalytic processes in the presence of artificial and solar illumination is an efficient method for the degradation of BPS, as well as its toxicity reduction. In most of these studies, the effects of different reaction conditions on the degradation of the pollutant were examined. It is noted that increasing the amount of catalyst or the catalyst surface area or increasing the H_2_O_2_ concentration or increasing in pH, in general, leads to an increased pollutants degradation efficiency rate. However, research is needed to determine the threshold values at which each of these parameters becomes detrimental to the degradation process.

There is a growing body of evidence showing that AOP processes can be efficient in removing BPs from aquatic environments with an efficiency rate ranging from partial (UV/Fenton system) to almost complete, or complete removal (UV/Persulfate and UV/ photocatalytic system). However, the mechanisms by which AOP processes enhance the biodegradability of BPs are insufficiently known and further research to elucidate such mechanisms as well as to more extensively evaluate the toxicity of the BPs degradation products would strengthen the toxicological findings. To achieve total mineralization, the systems would need significant improvement and optimization. Efforts should be made during the optimization of each reaction parameter to ensure that using the additional reactants, or causing other degradation reactions would not additionally overload the ecosystem.

## Figures and Tables

**Figure 1 molecules-27-01874-f001:**
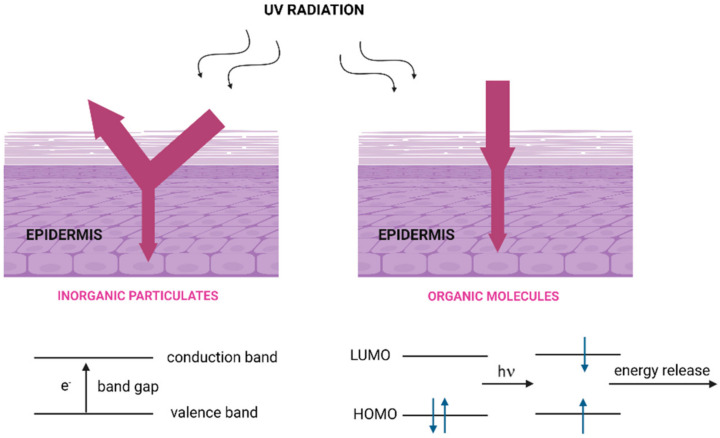
Method of action of inorganic (**left**) and organic (**right**) UV filters.

**Figure 2 molecules-27-01874-f002:**
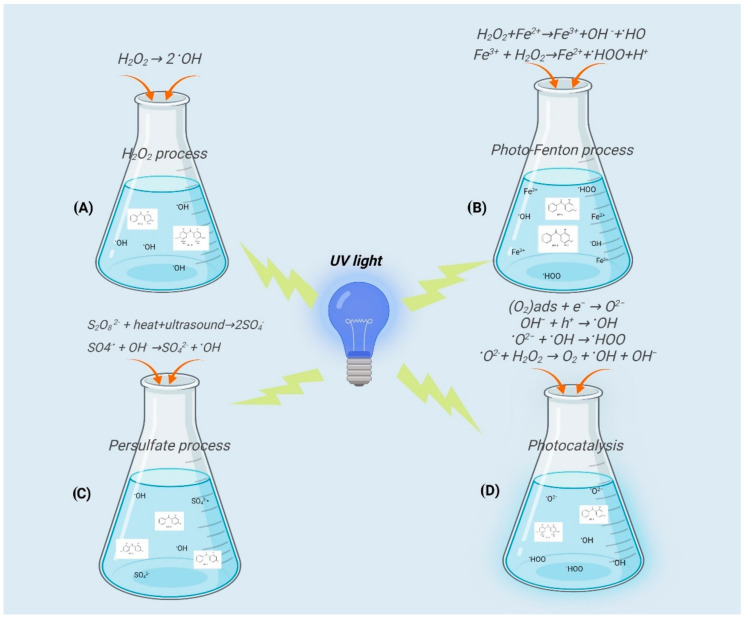
Common advanced oxidation processes for removal of benzophenone-type UV filters from water compartment.

**Figure 3 molecules-27-01874-f003:**
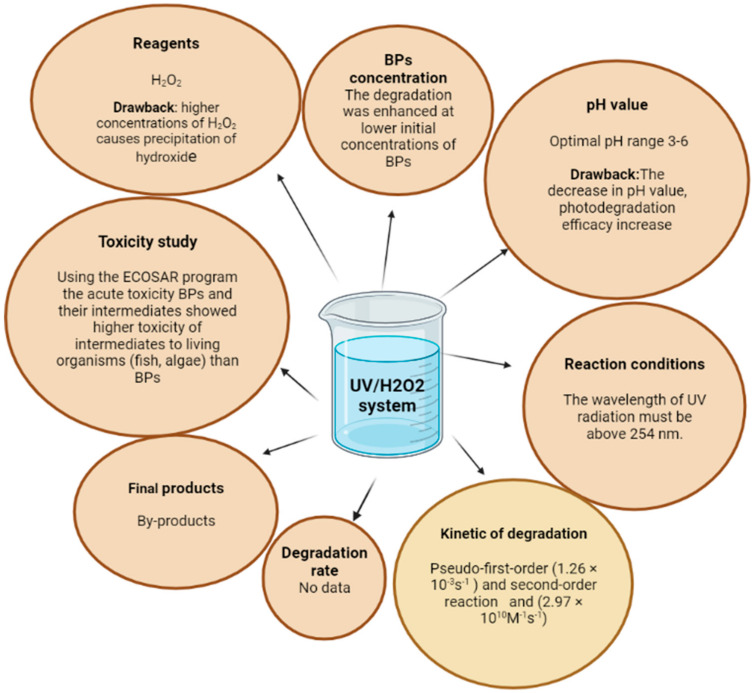
An overview of the effect of different conditions in the UV/H_2_O_2_ system on the degradation process of BPs. Created in BioRender.com.

**Figure 4 molecules-27-01874-f004:**
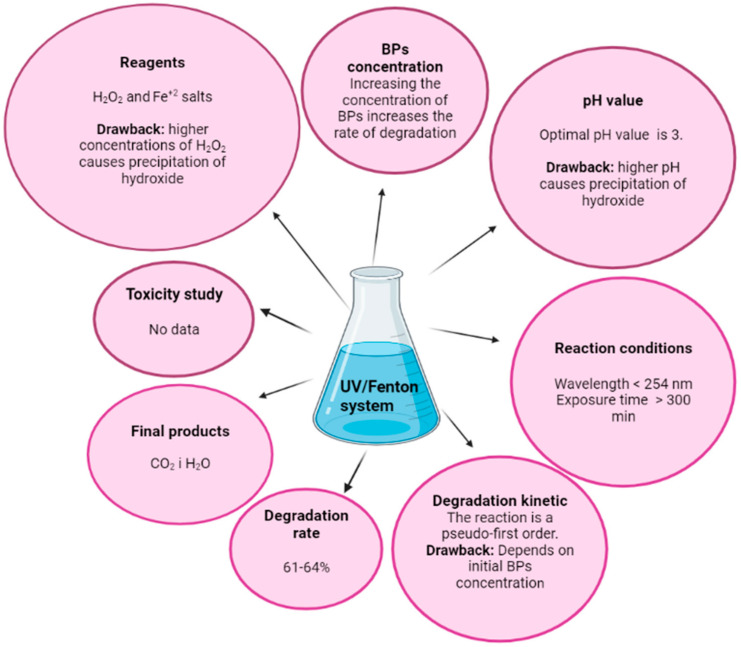
An overview of the effect of different conditions in the UV/Fenton process on the degradation process of BPs. Created in BioRender.com.

**Figure 5 molecules-27-01874-f005:**
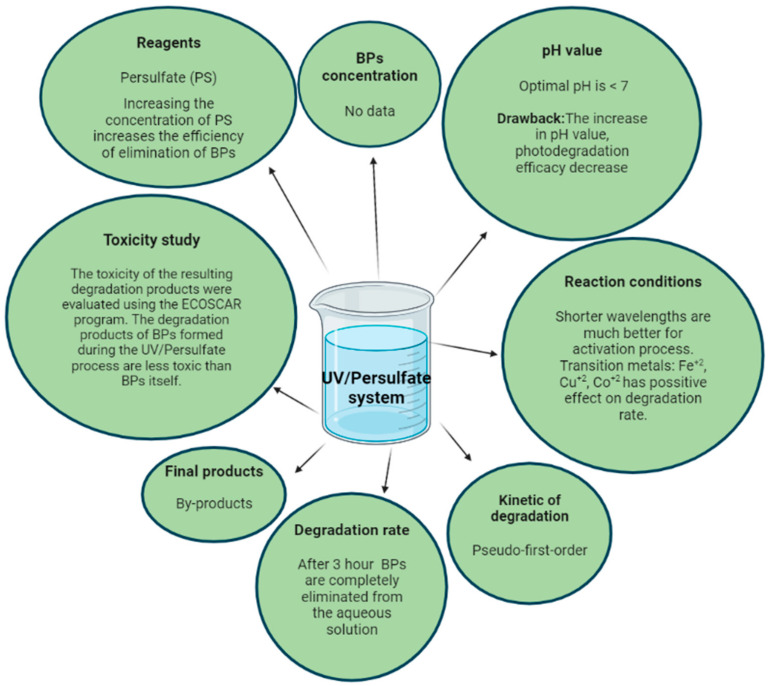
An overview of the effect of different conditions in the UV/persulfate process on the degradation process of BPs. Created in BioRender.com.

**Figure 6 molecules-27-01874-f006:**
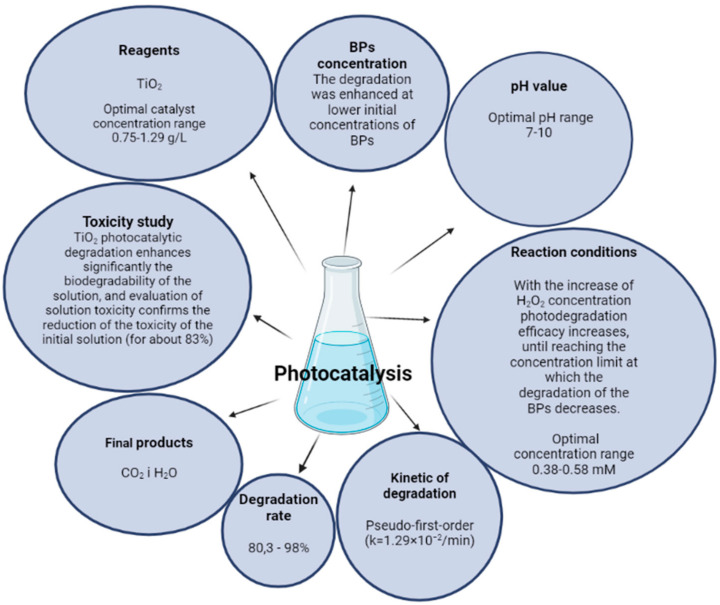
An overview of the effect of different conditions in the UV/TiO_2_ photocatalytic system on the degradation process of BPs. Created in BioRender.com.

**Table 1 molecules-27-01874-t001:** Classification of organic UV filters according to chemical structure.

Group	Typical Representatives
Benzophenone derivatives	Benzophenone-3 (BP3), benzophenone-4 (BP4)
p-Aminobenzoic acid and its derivatives (PABA)	Ethylhexyl dimethyl PABA (OD-PABA)
Dibenzoylmethane derivatives	4-tert-Butyl-47-methoxydibenzoylmethane (avobenzone)
Salycilates	Homosalate (HMS)
Methoxycinnamates	Ethylhexyl methoxycinnamate (OMC)
Camphor derivatives	4-methylbenzylidene camphor (4-MBC)
Triazine derivatives	Ethylhexyltriazone (OT)
Benzotriazole derivatives	Drometrizole trisiloxane (DRT)
Benzoimidazole derivatives	Phenylbenzimidazole sulfonic acid (PMDSA)
Others	Octocrylene (OCR)

**Table 2 molecules-27-01874-t002:** Structures and properties of benzophenone-type UV filters [16].

INCI Name ^1^	Abbreviation	Structure	Cas No. ^2^	Log Kow ^2^	Molecular Weight (g/mol)	Water Solubility (mg L−1) ^2^
Benzophenone	BP	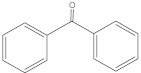	119-61-9	3.18	182.22	137
Benzophenone-1; 2,4-Dihydroxybenzophenone	BP-1	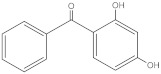	131-56-6	2.96	214.22	413.4
Benzophenone-2; 2,2’,4,4’-tetrahydroxybenzophenone	BP-2	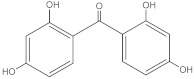	131-55-5	2.78	246.22	399
Benzophenone-3; 2-hydroxy-4-methoxybenzophenone	BP-3	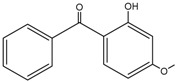	131-57-7	3.79	228.24	68.56
Benzophenone-4; 2-hydroxy-4-methoxybenzophenone-5-sulfonic acid	BP-4	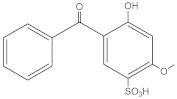	4065-45-6	0.37	308.31	2.03 × 104
Benzophenone-5; 2-hydroxy-4-methoxybenzophenone-5-sodium sulfonate	BP-5	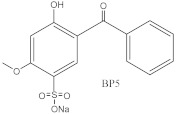	6628-37-1	−1.42	330.28	No data
Benzophenone-8; 2,2′-dihydroxy-4-methoxybenzophenone	BP-8	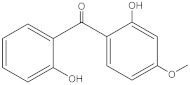	131-53-3	3.82	244.24	52.73
Benzophenone-10; 2-hydroxy-4-methoxy-4′-methylbenzophenone	BP-10	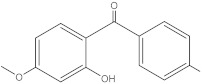	1641-17-4	4.07	242.27	33.03
Benzophenone-9; 2,2′-Dihydroxy-4,4′-dimethoxybenzophenone-5,5′-disulfonic acid disodium salt	BP-9	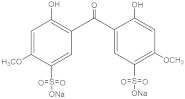	76656-36-5	−2.78	476.36	8.89 × 105
Benzophenone-12; 2-hydroxy-4-octyloxybenzophenone	BP-12	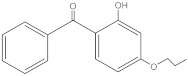	1843-05-6	6.96	326.18	0.037
2-hydroxybenzophenone	2HB	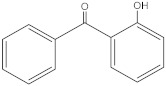	117-99-7	3.52	198.22	167.5
3-hydroxybenzophenone	3HB	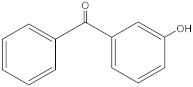	13020-57-0	2.67	198.22	896.5
4-hydroxybenzophenone	4HB	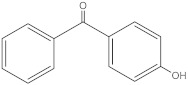	1137-42-4	3.07	198.22	406
4,4-Dihydroxybenzophenone	4HBP	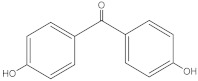	611-99-4	2.19	214.20	1.91 × 103
Diethylamino hydroxybenzoyl hexyl benzoate	DHHB	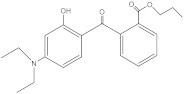	302776-68-7	6.54	397.51	8.2 × 10-3
2,3,4-trihydroxybenzophenone	234THB	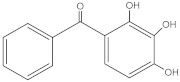	1143-72-2	2.91	230.22	381.1
4-phenylbenzophenone	4PB	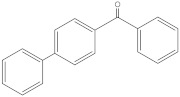	2128-93-0	4.91	258.314	1.36
2,2′-dihydroxybenzophenone	2DHB	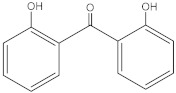	835-11-0	3.74	214.217	89.69
2,4,4′-trihydroxybenzophenone	244THB	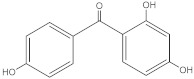	1470-79-7	2.48	230.216	837.4

Note: ^1^ INCI (International Nomenclature for Cosmetic Ingredient) elaborated by CTFA and Cosmetic Europe (former COLIPA). Note: ^2^ Source: ChemSpider website: http://www.chemspider.com, accessed on 10 January 2022.

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
