# Peer review of "Stability and Removal of Benzophenone-Type UV Filters from Water Matrices by Advanced Oxidation Processes"

_molecules, 2022, doi:10.3390/molecules27061874_

Round 1

Reviewer 1 Report

The authors revised most of the questions according to their comments, but two questions remained unsolved.

(1) In the section "2. Photodegradation of benzophenone-type UV filters in aquatic environments" and "3.4.2. Photocatalytic degradation of benzophenone-type UV filters", there are too many small paragraphs without logic and hierarchy, so why not divide them into subheading by the category description?

(2) The literatures cited is of poor timeliness, and more recent reports need to be reviewed.

(3) The illustrations used in the article are text combinations, and more high-quality original research drawings should be used.

Author Response

Reviewer 1

Thank you for taking the time to read this manuscript. Your feedback is appreciated.

  • In the section "2. Photodegradation of benzophenone-type UV filters in aquatic environments" and "3.4.2. Photocatalytic degradation of benzophenone-type UV filters", there are too many small paragraphs without logic and hierarchy, so why not divide them into subheading by the category description?

Section 2 and section 3.4.2. we divided into subheading (see manuscript)

  • The literatures cited is of poor timeliness, and more recent reports need to be reviewed.

As reviewers suggested, we researched the literature once again after the first revision round in February 2022 when we updated the cited literature with those references which are specifically relevant to the BPs and their stability and removal from water matrices. We did the literature search one more time by March 10th  but we didn’t find any additional literature that is relevant to our work.

  • The illustrations used in the article are text combinations, and more high-quality original research drawings should be used.

The authors' intention was to graphically present the main elements of the processes which are in detail described in the text of the manuscript. Our intention was for the reader to get basic information about each process based on them in a simple way. From that reason we would like to keep them as such.

Reviewer 2 Report

In the current state, the paper is available for publication in this journal.

Author Response

Thank you for taking the time to read this manuscript. Your feedback is appreciated.

This manuscript is a resubmission of an earlier submission. The following is a list of the peer review reports and author responses from that submission.

Round 1

Reviewer 1 Report

The authors reviewed the degradation kinetics and mechanisms of benzophenone-type UV filters and their degradation products (DPs) under UV and solar irradiation and in UV-based advanced oxidation processes (AOPs) like: UV/H2O2, UV/persulfate and Fenton process. However, the content description of the review is illogical and unreadable. Acceptance of the review can be reconsidered after substantial revisions:

  1. Some necessary diagrams must be present in the review, including an organization chart for the review, as well as some representative figure form published research papers for each chapter, such as “Schematic diagram of photodegradation of benzophenone type UV filter in aquatic environment”, “Schematic diagram of advanced oxidation processes”, “Schematic diagram of degradation kinetics and mechanisms” and so on. Each chapter must have some figures to support the description of the essay.
  2. In the second chapter, there are too many small paragraphs without logic and hierarchy, so why not divide them into SubTitle by the category description? The same question comes up many times in the text. If necessary, tertiary headings should also be employed.
  3. There are too few summaries of photocatalytic reactions used s in degradation reaction of BPs, some of the latest research needs to be added.
  4. The Conclusion of chapter 4 should be divided into Conclusion and Prospect. In particular, the section of Prospect is very important for the review.
  5. Photocatalysis is a good technology for the treatment of water pollution by nanomaterials, some reports can be cited: “Journal of Hazardous Materials, 2021, 127941”, “Journal of Materials Chemistry A, 2020, 8, 13038” and “Chemical Engineering Journal, 2020, 389, 123431.”.

Reviewer 2 Report

In this review article, the authors mainly summarize the degradation of benzophenone-type UV filters and their degradation products under UV and solar irradiation and in UV-based advanced oxidation processes like: UV/H2O2, UV/persulfate and Fenton process. In my opinion, the authors present only the results of some references, which is insufficient from the view point of novelty. In addition, the manuscript is expression trival and unsatisfactory that should be presented more concise way. Authors should also consider to adjust the structure of the article and add some reviews. Moreover, the reviewer’s passion is really turned off by the poor writing skills and many grammar errors in the manuscript. The reviewer wonders if the manuscript had been polished and carefully checked before submitting to this journal. Therefore, Therefore, I do not think this work is strong enough to be published on Molecules. The main concerns are as follows:

>1. The authors should carefully correct the language errors in the manuscript, include the grammar, word use, and punctuation.

>2. The review article mainly discuss the degradation of benzophenone-type UV filters under different technologies, so defining the title would make the review article more informative.

>3. In order to make the article more organized, the 2 and 3.4 parts should be classified as together to study the photodegradation of benzophenone-type UV filters. This is mainly because advanced oxidation processes contain photocatalytic process.

>4. In order to make this article more scientific, the authors should add some figures to state the results, especially for degradation mechanism of benzophenone-type UV filters.
